# QUANTIZED SPIKE-DRIVEN TRANSFORMER

**Xuerui Qiu**[1,2,3][*]**Malu Zhang**[1][†]**Jieyuan Zhang**[1][*], **Wenjie Wei**[1], **Honglin Cao**[1],
**Junsheng Guo**[4], **Rui-Jie Zhu**[5], **Yimeng Shan**[6], **Yang Yang**[1], **Haizhou Li**[7]

[1]University of Electronic Science and Technology of China,
[2]Institute of Automation, Chinese Academy of Sciences,
[3]School of Future Technology, University of Chinese Academy of Sciences,
[4]China Agricultural University, [5]University of California, Santa Cruz,
[6]Liaoning Technical University, [7]Chinese University of Hong Kong (Shenzhen)

## ABSTRACT

Spiking neural networks (SNNs) are emerging as a promising energy-efficient alternative to traditional artificial neural networks (ANNs) due to their spike-driven paradigm. However, recent research in the SNN domain has mainly focused on enhancing accuracy by designing large-scale Transformer structures, which typically rely on substantial computational resources, limiting their deployment on resource-constrained devices. To overcome this challenge, we propose a quantized spike-driven Transformer baseline (QSD-Transformer), which achieves reduced resource demands by utilizing a low bit-width parameter. Regrettably, the QSD-Transformer often suffers from severe performance degradation. In this paper, we first conduct empirical analysis and find that the bimodal distribution of quantized spike-driven self-attention (Q-SDSA) leads to spike information distortion (SID) during quantization, causing significant performance degradation. To mitigate this issue, we take inspiration from mutual information entropy and propose a bi-level optimization strategy to rectify the information distribution in Q-SDSA. Specifically, at the lower level, we introduce an information-enhanced LIF to rectify the information distribution in Q-SDSA. At the upper level, we propose a fine-grained distillation scheme for the QSD-Transformer to align the distribution in Q-SDSA with that in the counterpart ANN. By integrating the bi-level optimization strategy, the QSD-Transformer can attain enhanced energy efficiency without sacrificing its high-performance advantage. We validate the QSD-Transformer on various visual tasks, and experimental results indicate that our method achieves state-of-the-art results in the SNN domain. For instance, when compared to the prior SNN benchmark on ImageNet, the QSD-Transformer achieves 80.3% top-1 accuracy, accompanied by significant reductions of $6.0\times$ and $8.1\times$ in power consumption and model size, respectively. Code is available at Quantized Spike-driven Transformer.

## 1 INTRODUCTION

Spiking neural networks (SNNs) have emerged as a promising approach for realizing energy-efficient computational intelligence due to their high biological plausibility (Maass, 1997), sparse spike-driven communication (Roy et al., 2019), and low power consumption on neuromorphic hardware (Davies et al., 2018; Pei et al., 2019; Merolla et al., 2014). Within SNNs, the spiking neuron transmits information via sparse binary spikes, where the binary value of 0 denotes neural quiescence and the value of 1 signifies a spiking event (Shrestha & Orchard, 2018; Eshraghian et al., 2023). The unique spike-driven nature is key to achieving low power consumption, where only a subset of spiking neurons are activated to perform sparse synaptic accumulation (AC) (Yao et al., 2023a;b). However, despite their high energy efficiency, the application of SNNs is constrained by their low task accuracy.

---

[*]Equal Contribution
[†]Corresponding author, maluzhang@uestc.edu.cn

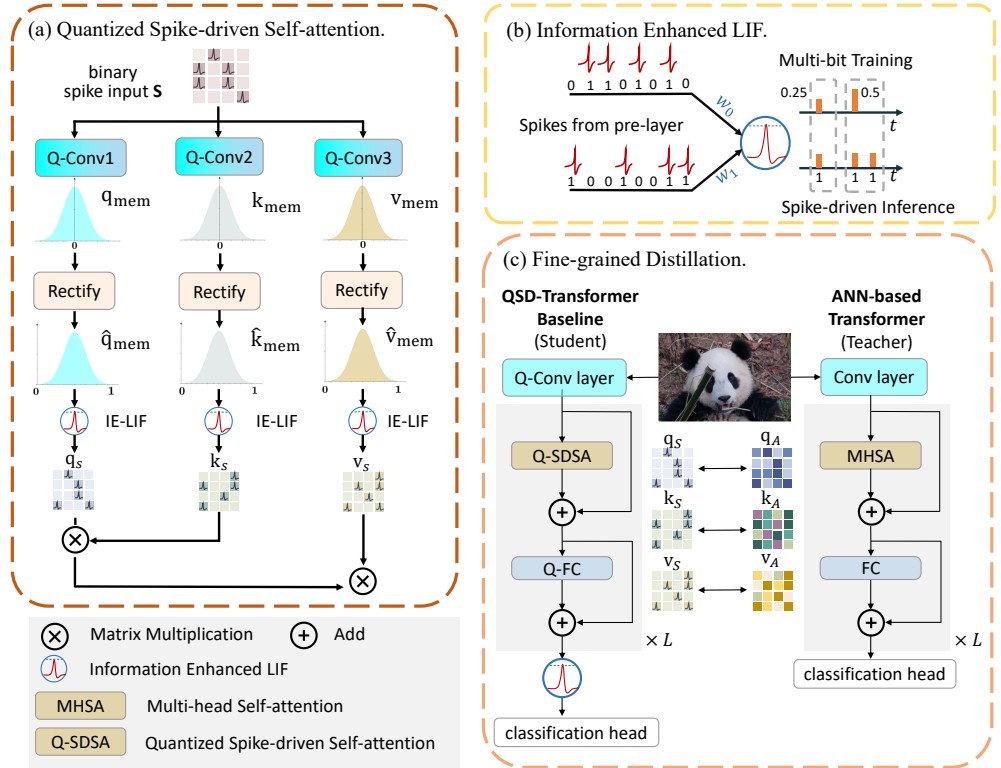

Figure 1: Overview of the QSD-Transformer. (a) Proposed quantized spike-driven self-attention (Q-SDSA) module, where the membrane potential is rectified and then sent to the information-enhanced LIF (IE-LIF) neuron. (b) Proposed IE-LIF spiking neuron model, which utilizes the multi-bit spike during training while the binary spike during inference. (c) Proposed fine-grained distillation scheme.

Numerous researchers have made great efforts to improve the performance and expand the application scenarios of SNNs. Building upon the success of Vision Transformers (ViT) (Dosovitskiy et al., 2020; Touvron et al., 2021; Yu et al., 2023), researchers naturally combined SNNs with Transformers, resulting in significant performance improvements on ImageNet benchmark (Zhou et al., 2023b;a; 2024b;a) and diverse application scenarios (Zhang et al., 2022b;c; Lv et al., 2023). Despite their commendable performance, these studies come at the cost of massive model parameters and high computational complexity. For instance, Spikformer v2 (Zhou et al., 2024c) and Spike-driven Transformer v3 Yao et al. (2024) achieve accuracies of 82.4% and 86.2% on the ImageNet dataset, respectively. These models have 173M parameters, necessitating 1384MB memory, and requiring 28.4G synapse-operations per second for inference. This places significant demands on the storage and computational capabilities of neuromorphic chips, thereby limiting their deployment on edge devices. Therefore, there is an urgent need for a low-bit and high-performance Spike-based Transformer.

Numerous efforts have been made to compress and accelerate neural networks on edge computing devices, e.g., pruning (Han et al., 2015; Shen et al., 2023), quantization (Qin et al., 2021; Deng et al., 2023), and knowledge distillation (Hinton et al., 2015; Xu et al., 2023). Among these, quantization is particularly suitable for hardware deployment as it can reduce the bit-width of network parameters and activations, enabling efficient inference. The post-training quantization (PTQ) approach (Jacob et al., 2018) calculates quantization parameters directly based on pre-trained full-precision models, which may limit the model's performance to a suboptimal level without fine-tuning. In particular, the model obtained from this approach may suffer from dramatic performance drops when quantized to ultra-low bits (e.g., 2, 4 bit). In contrast to PTQ, Quantization-Aware Training (QAT) (Krishnamoorthi, 2018) performs quantization during the learning process and generally achieves great performance with high compression ratios. However, in the field of SNNs, research on QAT methods has primarily focused on convolutional neural networks (CNNs), with low-bit Spikformer remaining largely unexplored.

In this paper, we first construct a quantized spike-driven Transformer (QSD-Transformer) baseline (Yao et al., 2023a), which directly quantizes 32-bit weights to low bit-width during training. Despite exhibiting significant energy efficiency, this simple method can lead to severe performance degradation. Through detailed analysis of the baseline, we reveal that quantizing the attention module will reduce the representation capability of the self-attention maps, which is defined as the spiking information distortion (SID) problem. This is the main reason for the performance degradation. To address this issue, we propose a bi-level optimization strategy for the baseline, aiming at rectifying the distribution in quantized spike-driven self-attention maps (Q-SDSA) from both the neuron and network levels. The overview of the QSD-Transformer is shown in Fig. 1 and our main contributions can be summarized as:

- We construct a lightweight spike-driven Transformer baseline through quantization, called QSD-Transformer. The QSD-Transformer quantizes the synaptic weights from a 32-bit to a low-bit representation (typically 2, 3, and, 4 bits), leading to reduced model size and significant energy efficiency advantages.

- We reveal that the proposed baseline suffers from performance degradation due to the SID problem in Q-SDSA. Inspired by information entropy, we propose a bi-level optimization strategy to solve this issue. This strategy introduces an information-enhanced LIF and a fine-grained distillation to rectify the distribution of Q-SDSA, leading to enhanced performance.

- We validate the QSD-Transformer on various visual tasks, e.g., classification, object detection, semantic segmentation, and transfer learning. Experimental results indicate that our method outperforms existing spiking Vision Transformers by a substantial margin, while also boasting a compact model size and extremely low power consumption.

## 2 RELATED WORKS

**Spiking vision transformer.** Spikformer (Zhou et al., 2023b) pioneered direct training with a pure SNN architecture, introducing a linear self-attention mechanism that eliminates multiplication by activating Query, Key, and Value with spiking neurons and replacing softmax with spiking neurons. Its successor (Zhou et al., 2024c) integrated masked image modeling (He et al., 2022), achieving an 82.25% accuracy on ImageNet with 172 M parameters, the highest among SNNs. SpikingResformer (Shi et al., 2024) introduces a novel spike self-attention mechanism along with a judicious scaling approach, enabling effective extraction of local features. However, none of these models preserved the spike-driven nature until the spike-driven Transformer (Yao et al., 2023b), which introduces the sparse addition to self-attention using only masking operations. Its successor (Yao et al., 2023a) focused on the meta-design of the spiking vision Transformer, including architecture, spike-driven self-attention, shortcut connections, etc. The proposed spike-driven Transformer v2 (Yao et al., 2023a) set up direct training SNN backbone for improving performance across tasks like image classification, segmentation, and object detection, hinting at impacts on neuromorphic chip design. Hence, in this study, we adopt the pure addition spike-driven Transformer v2 for quantization baseline.

**Model compression.** Numerous compression techniques have been explored to compress large-scale SNNs, including: (1) Pruning (Han et al., 2015; Kusupati et al., 2020; Savarese et al., 2020) in SNNs generally draw on traditional pruning methods from ANNs to suit the spatial and temporal domains (Chen et al., 2022; Shi et al., 2023; Shen et al., 2024). While successful on simpler datasets and shallow networks, achieving high performance becomes more challenging with complex datasets and deeper networks. (2) Knowledge distillation (Hinton et al., 2015; Guo et al., 2023; Touvron et al., 2021) involves the transfer of knowledge from large-scale ANNs or SNNs to smaller-scale SNNs, aiming to compress models and reduce energy consumption. However, these methods (Takuya et al., 2021; Tran et al., 2022; Xu et al., 2023) often distill only the final output of the model, leading to incomplete knowledge transfer and suboptimal performance in SNNs. (3) Quantization (Jacob et al., 2018; Krishnamoorthi, 2018), particularly for hardware deployment, is advantageous as it reduces the bit-width of network parameters and activations, enabling efficient inference. Recent research on quantization methods (Stromatias et al., 2015; Deng et al., 2023; Kheradpisheh et al., 2022) for SNNs has predominantly focused on weight binarization within Conv-based architectures. For instance, Deng et al. (Deng et al., 2023) utilized QAT (Krishnamoorthi, 2018; Jacob et al., 2018) to reduce the weight size of Conv-based SNN, which demonstrated high compression performance with acceptable accuracy loss on recognition tasks. Despite the significant potential of QAT in reducing the

memory and computational costs of Conv-based SNN (Deng et al., 2023), directly applying QAT on Spikformer leads to poor performance. The core challenge is the significant distribution discrepancy between the binary spike patterns and the normal distribution in the self-attention of Spikformer and ANN Transformer, which causes severe information distortion, leading to performance degradation.

## 3 PRELIMINARY

In this section, we first introduce the spiking neuron model. Then, we construct a quantized spike-driven Transformer (QSD-Transformer) baseline, which quantizes the synaptic weight from 32-bit to low bit-width, thereby demonstrating significant energy efficiency advantages.

**Spiking neuron model.** In this paper, we choose the widely-employed iterative Leaky Integrate-and-Fire (LIF) model (Wu et al., 2018; Guo et al., 2024), which can be described by the following mathematical equations:

$$\mathbf{v}^\ell[t] = \mathbf{h}^\ell[t-1] + f(\mathbf{w}^\ell, \mathbf{x}^{\ell-1}[t-1]), \tag{1}$$

$$\mathbf{s}^\ell[t] = \boldsymbol{\Theta}(\mathbf{v}^\ell[t] - \vartheta), \tag{2}$$

$$\mathbf{h}^\ell[t] = \tau \mathbf{v}^\ell[t] - \mathbf{s}^\ell[t], \tag{3}$$

where $\tau$ is the time constant, $t$ is the time step, $\mathbf{w}^\ell$ is the weight matrix of layer $\ell$, $f(\cdot)$ is the operation that stands for convolution (Conv) or fully connected (FC), $\mathbf{x}$ is the input, and $\boldsymbol{\Theta}(\cdot)$ denotes the Heaviside step function. When the membrane potential $\mathbf{v}$ exceeds the firing threshold $\vartheta$, the LIF neuron will trigger a spike $\mathbf{s}$; otherwise, it remains inactive. After spike emission, the neuron invokes the reset mechanism, where the soft reset function is employed. $\mathbf{h}$ is the membrane potential following the reset function.

**QSD-Transformer baseline.** We select the purely spike-driven Transformer v2 (SD-Transformer v2) (Yao et al., 2023a) to perform quantization, and the LSQ (Bhalgat et al., 2020) method is employed to quantize the 32-bit weights to low bit-width (e.g., 2, 3, 4 bits). The quantization function is defined as:

$$\mathcal{Q}(\mathbf{w}) = \left\lfloor \text{clip}\left\{ \frac{\mathbf{w}}{\alpha_\mathbf{w}}, -2^{b-1}, 2^{b-1}-1 \right\} \right\rceil, \quad \hat{\mathbf{w}} = \alpha_\mathbf{w} \mathcal{Q}(\mathbf{w}), \tag{4}$$

where $\mathbf{w}$ is the 32-bit weight, $b$ is the bit assigned to the quantized weight (i.e., $\mathcal{Q}(\mathbf{w})$), and $\alpha_\mathbf{w}$ is the scaling factor used to mitigate the quantization error. Moreover, $\text{clip}\{x, a, b\}$ confines $x$ within range $[a, b]$, and $\lfloor \cdot \rceil$ denotes the rounding operator. These two operations make the quantization function non-differentiable, so we adopt the straight-through estimator (STE) (Bengio et al., 2013) to assist the gradient backpropagation. Eq. 4 is performed for all weight layers in the baseline model. Building upon this, the calculation for a certain layer $\ell \in \{\text{FC}, \text{Conv}\}$ in our baseline is expressed as:

$$\mathcal{Q}_\ell(\mathbf{x}) = \hat{\mathbf{w}} \cdot \mathcal{SN}(\mathbf{x}) = \alpha_\mathbf{w} \mathcal{Q}(\mathbf{w}) \cdot \mathcal{SN}(\mathbf{x}). \tag{5}$$

Here, $\mathcal{SN}(\cdot)$ represents the spiking neuron layer, which converts the floating-point input $\mathbf{x}$ into the binary spike. Hence, the QSD-Transformer employs binary spike activities and low bit-width weights for Conv and FC operations. This replaces the original computationally intensive operations, leading to significant energy efficiency improvements. Following Eq. 5, the quantization for the spike-driven self-attention (Q-SDSA) can be further described as:

$$\mathbf{q} = \mathcal{Q}_{\text{Conv1}}(\mathbf{x}), \mathbf{k} = \mathcal{Q}_{\text{Conv2}}(\mathbf{x}), \mathbf{v} = \mathcal{Q}_{\text{Conv3}}(\mathbf{x}), \quad \text{Q-SDSA}(\mathbf{q}, \mathbf{k}, \mathbf{v}) = \mathcal{SN}((\mathbf{q_s}\mathbf{k_s}^T)\mathbf{v_s}), \tag{6}$$

where, $\mathbf{q_s} = \mathcal{SN}(\mathbf{q})$, and $\mathbf{k_s}$, $\mathbf{v_s}$ are obtained in the same way. It can be observed from Eq. 6 that our Q-SDSA module reduces the computational number by the linear attention mechanism with $\mathcal{O}(ND^2)$, where $N$ is the token numbers and $D$ is the channel dimensions. However, quantizing the attention module will diminish its representation capacity, leading to performance degradation. In the next section, we will provide a detailed explanation of this issue and propose methods to address it.

## 4 METHOD

In this section, we first reveal that the performance degradation of our baseline is due to the limited representational capacity of Q-SDSA. Inspired by information entropy theory, we propose a bi-level

optimization strategy to address this issue. At the lower level, we introduce the information-enhanced leaky integrate-and-fire (IE-LIF) neuron, which maximizes information entropy by adjusting the spike distribution. At the upper level, we present the fine-grained distillation scheme, which minimizes conditional entropy by aligning the information of Q-SDSA with that of ANNs.

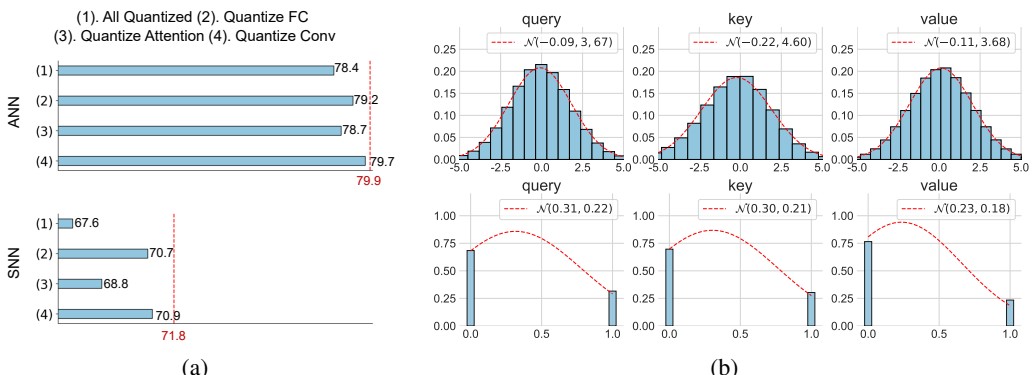

Figure 2: (a) Accuracy of quantizing different modules in the SD-Transformer v2 and its same ANN Transformer. (b) The distribution of the attention module (blue shadow), and the probability density function curve of normal distribution (red line). Experiments are conducted on ImageNet, and 3 layers in SD-Transformer v2 and its same ANN Transformer are selected for illustration.

## 4.1 PERFORMANCE DEGRADATION ANALYSIS

Despite its efficiency advantages, the QSD-Transformer baseline suffers from performance degradation, as shown in Fig. 2 (a) (All quantized). In contrast, the quantized ANN Transformer can balance both efficiency and performance (Wu et al., 2022). Hence, we quantize each module of the architecture, i.e., FC, Conv, and Attention, to identify which one has the biggest impact on performance degradation.

We illustrate the ablation results in Fig. 2 (a), where all weights are quantized to 4-bit. Obviously, the attention layer in SNNs is highly sensitive to quantization, but it is not observed in ANNs. This is attributed to the different information distributions in the self-attention map, as depicted in Fig. 2 (b). It can be observed that the information within Q-SDSA displays a bimodal distribution, whereas the information in ANN adheres to a normal distribution. Due to the utilization of both low-bit weights and binary spikes, the information representation capability of the Q-SDSA is severely limited compared to that of ANNs. We define it as the spike information distortion (SID) problem.

To solve the SID issue, we draw on the quantized Transformer in the ANN domain (Liu et al., 2021) that has struck a good balance between efficiency and performance by maintaining the noraml distribution of activity. This prompts us to adjust the information distribution in the Q-SDSA to match that of ANNs. To achieve this, we take inspiration from the information entropy theory (Paninski, 2003) and formulate it as the mutual information entropy maximization problem.

**Definition 1.** *Addressing the performance degradation of the QSD-Transformer baseline is equivalent to maximizing the mutual information entropy between it and the quantized Transformer in ANNs. The optimization goal for the QSD-Transformer is defined below.*

$$\max_{\theta^{\mathcal{S}}} \mathcal{I}(\mathbf{p}^{\mathcal{S}}; \mathbf{p}^{\mathcal{A}}) = \mathcal{H}(\mathbf{p}^{\mathcal{S}}) - \mathcal{H}(\mathbf{p}^{\mathcal{S}}|\mathbf{p}^{\mathcal{A}}), \tag{7}$$

where $\mathbf{p}^{\mathcal{S}}$ and $\mathbf{p}^{\mathcal{A}}$ are the attention score value in SNN and ANN respectively, and $\theta^{\mathcal{S}}$ is the parameters of the QSD-Transformer. However, directly optimizing Eq. 7 is challenging, so we regard it as a bi-level optimization problem (Colson et al., 2007; Sinha et al., 2017). It is achieved by minimizing the conditional information entropy $\mathcal{H}(\mathbf{p}^{\mathcal{S}}|\mathbf{p}^{\mathcal{A}})$ and maximizing the information entropy $\mathcal{H}(\mathbf{p}^{\mathcal{S}})$, which is defined as:

$$\min_{\theta^{\mathcal{S}}} \mathcal{H}(\mathbf{p}^{\mathcal{S}^{\star}}|\mathbf{p}^{\mathcal{A}}), \quad \text{s.t.} \quad \mathbf{p}^{\mathcal{S}^{\star}} = \arg\max_{\mathbf{p}^{\mathcal{S}}} \mathcal{H}(\mathbf{p}^{\mathcal{S}}). \tag{8}$$

To accomplish it, we first propose the information-enhanced LIF (IE-LIF) neuron, aiming to maximize the information entropy $\mathbf{p}^{\mathcal{S}*}$ at the lower level. We further introduce a fine-grained distillation (FGD) scheme, aiming to minimize the conditional entropy $\mathcal{H}(\mathbf{p}^{\mathcal{S}}|\mathbf{p}^{\mathcal{A}})$ at the upper level.

## 4.2 INFORMATION-ENHANCED LIF NEURON

As mentioned in Fig. 2 (b), the information in the self-attention map of the QSD-Transformer follows a binomial distribution, which limits the representational capacity of the attention module. Therefore, we propose the information-enhanced LIF (IE-LIF) neuron and adjust the information distribution of Q-SDSA at the lower level, focusing on maximizing the information entropy $\mathcal{H}(\mathbf{p}^{\mathcal{S}})$.

**Proposition 1.** *Given the SNN Transformer and ANN Transformer models, where the distributions of the query ($\mathbf{q}$), key ($\mathbf{k}$), and value ($\mathbf{v}$) follow binomial $\mathcal{B}(r, T)$ and normal $\mathcal{N}(\mu, \sigma)$ distributions, respectively, it is postulated that as the SNN's time step $T$ tends to infinity, there exist parameters $\mu$, $\sigma$, and $r$ such that the average entropy over time of the SNN's attention scores $\mathcal{H}(\sum_{t=1}^{T} \mathbf{p}^{\mathcal{S}}[t])$ equals ANN attention scores' entropy $\mathcal{H}(\mathbf{p}^{\mathcal{A}})$.*

Proof can be found in Appendix B. According to Proposition 1., within infinite time steps $T$, the attention matrix values (e.g., $\mathbf{q_s}, \mathbf{k_s v_s}\}$) in the QSD-Transformer have the same information representation as those in the ANN Transformer. However, numerous time steps $T$ will inevitably lead to latency and huge energy consumption. Recently, Hao et al. (2023b) and Hu et al. (2023) achieved high-performance conversion by transforming quantized ANNs into SNNs and using fewer time steps. This prompted us to train directly using multi-bit values. We only need to ensure that inference is spike-driven. Thus, we propose the concept of IE-LIF, in which Eq. 2 can be written as:

$$\mathbf{a}^{\ell}[t] = \frac{1}{b} \left\lfloor \text{clip}\{\mathbf{v}^{\ell}[t], 0, b\} \right\rfloor, \tag{9}$$

where $\mathbf{a}^{\ell}[t]$ is the multi-bit output of IE-LIF, and $b$ represents the maximum integer value emitted by the IE-LIF, which is equipped with the baseline. Since Eq. 9 is non-differentiable, we employ the straight-through estimator (STE) (Bengio et al., 2013) to retain the gradient derivation during backpropagation.

Previous SNNs have utilized multi-bit spikes (integers) (Hao et al., 2024; Ponghiran & Roy, 2022) or continuous values (Wu et al., 2021) to reduce quantization error, thereby alleviating the shortcomings of binary spikes. However, this approach raises concerns because it can undermine the inherent spike-driven characteristics of SNNs. We propose a solution where IE-LIF uses multi-bit values during the training phase and subsequently converts these values to binary spikes for inference, as depicted in Fig. 1 (c). Moreover, the output $\mathbf{x}^{\ell+1}[t]$ of each layer in the SNN is represented as:

$$\mathbf{x}^{\ell+1}[t] = \mathbf{w}^{\ell} \cdot \mathbf{a}^{\ell}[t] = \mathbf{w}^{\ell} \cdot \sum_{t=1}^{T} \mathbf{s}^{\ell}[t], \tag{10}$$

where $a^{\ell}[t]$ represents multi-bit spikes during training with one timestep and is denoted as $\{0, 0.25, 0.5, 0.75, 1\}^{T=1}$, while $s^{\ell}[t]$ represents binary spikes during inference and is extended to 4 virtual timesteps denoted as $\{0, 1\}^{T=4}$, with a maximum integer value $b$ set to 4 in this paper.

With the introduction of IE-LIF, our next step involves low-level optimization to maximize the entropy of attention scores $\mathbf{p}^{\mathcal{S}}$. We first observed that the membrane potentials $\{\mathbf{q}_{\text{mem}}, \mathbf{k}_{\text{mem}}, \mathbf{v}_{\text{mem}}\}$ in the IE-LIF model within Q-SDSA approximately follow a normal distribution $\mathcal{N}(\mu, \sigma)$ with $\mu = 0$, as also noted in (Guo et al., 2022a;b). Then we provide the formula for calculating the maximum information entropy.

**Proposition 2.** *Given a random variable $\mathbf{x}$ following a normal distribution $\mathcal{N}(\mu, \sigma)$, the information entropy $\mathcal{H}(\mathbf{x})$ achieves its maximum value of $\frac{1}{2} \log 2\pi e \sigma^2(\mathbf{x})$.*

Proof can be found in Appendix C. According to Proposition 2., the maximum information entropy of membrane potential is $\mathcal{H}(\mathbf{p}_{\text{mem}}) = \frac{1}{2} \log 2\pi e \sigma^2(\mathbf{p}_{\text{mem}})$, acting as the upper limit for the information entropy of spike attention scores. Through the application of IE-LIF, the information entropy of spike attention scores becomes a discrete representation of membrane potential's information entropy:

$$\mathcal{H}(\mathbf{p}^{\mathcal{S}}) = -\sum_{k=0}^{b} \left( G(\mathbf{p}_{\text{mem}})\delta(\mathbf{p}_{\text{mem}} - \frac{k}{b}) \right) \cdot \log \left( G(\mathbf{p}_{\text{mem}})\delta(\mathbf{p}_{\text{mem}} - \frac{k}{b}) \right), \tag{11}$$

where $G(\mathbf{p}_{\text{mem}})$ is the Gaussian distribution function of membrane potential $\mathbf{p}_{\text{mem}}$ and $\delta(\cdot)$ is the Dirac delta function and $\int_{-\infty}^{\infty} \delta(x)\, dx = 1$. However, since $\mathbf{p}_{\text{mem}} \sim \mathcal{N}(0, \sigma)$, when Eq. 9 is applied to membrane potentials $\mathbf{p}_{\text{mem}}$, spikes are emitted only when $\mathbf{p}_{\text{mem}}$ exceeds 0. This may lead to the distributions of attention scores resembling the right tail of a discrete normal distribution, causing mismatched attention scores between SNNs and ANNs. Hence, we propose a membrane potential rectify function (MPRF) $\phi^{\ell}(\cdot)$, which can be expressed as:

$$\hat{\mathbf{p}}_{\text{mem}} = \phi^{\ell}(\mathbf{p}_{\text{mem}}) = \frac{\mathbf{p}_{\text{mem}} - \mu(\mathbf{p}_{\text{mem}})}{\sigma(\mathbf{p}_{\text{mem}})} \cdot \gamma + \alpha, \tag{12}$$

where $\mathbf{p}_{\text{mem}}$ and $\hat{\mathbf{p}}_{\text{mem}}$ represent the membrane potential of Q-SDSA before and after applying the MPRF, and $\gamma, \alpha$ are the learnable hyperparameters. The MPRF is only executed when $\ell \in$ Q-SDSA. After MPRF, attention score distribution aligns more closely with the desired normal distribution, reducing the mismatch between SNNs and ANNs. At this point, the information entropy of SNN attention scores is $\mathbf{p}^{\mathcal{S}^{\star}} = \sum_{k=0}^{b} G(\hat{\mathbf{p}}_{\text{mem}}) \delta(\hat{\mathbf{p}}_{\text{mem}} - \frac{k}{b})$. Furthermore, the MPRF can be incorporated into the weights $\mathbf{w}^{\ell}$ during inference, details of which can be found in Appendix D.

### 4.3 FINE-GRAINED DISTILLATION

The IE-LIF neuron has maximized the information entropy of $\mathcal{H}(\mathbf{p}^{\mathcal{S}})$. Building upon this, we achieve the optimization goal of Eq. 8 by proposing a fine-grained distillation (FGD), which adjusts the distribution of Q-SDSA at the upper level to minimize the conditional entropy $\mathcal{H}(\mathbf{p}^{\mathcal{S}^{\star}}|\mathbf{p}^{\mathcal{A}})$.

The proposed FGD achieves minimal conditional entropy by minimizing the norm distance between $\hat{\mathbf{p}}^{\mathcal{S}^{\star}}$ and $\hat{\mathbf{p}}^{\mathcal{A}}$, with the optimal solution being $\hat{\mathbf{p}}^{\mathcal{S}^{\star}} = \hat{\mathbf{p}}^{\mathcal{A}}$. It utilizes appropriate distillation activations and meticulously designed similarity matrices to effectively leverage knowledge from the teacher model. Therefore, the FGD scheme is defined as:

$$\mathcal{L}_{\text{FGD}} = \sum_{\mathbf{p} \in \{\mathbf{q}, \mathbf{k}, \mathbf{v}\}} \sum_{l=1}^{L} \sum_{h=1}^{H} ||\mathcal{F}_{\mathbf{p}^{\mathcal{A}}}^{l} - \mathcal{F}_{\mathbf{p}^{\mathcal{S}}}^{l}||, \quad \text{where} \quad \mathcal{F}_{\mathbf{p}} = \frac{\mathbf{p} \times \mathbf{p}^{\top}}{||\mathbf{p} \times \mathbf{p}^{\top}||}, \tag{13}$$

where $L$ denotes the number of layers in the Transformer, $H$ represents the number of heads, and $||\cdot||$ indicates $\ell_2$ normalization. During backpropagation, gradient updation drives the attention matrix of the QSD-Transformer and the same ANN Transformer closer, thereby minimizing $\mathcal{H}(\mathbf{p}^{\mathcal{S}^{\star}}|\mathbf{p}^{\mathcal{A}})$. The overall training loss function $\mathcal{L}$ of our QSD-Transformer is defined as:

$$\mathcal{L} = \mathcal{L}_{\text{CE}}(\sum_{t=1}^{T} \mathbf{s}^{\ell}[t], y) + \lambda \mathcal{L}_{\text{FGD}}, \tag{14}$$

where $\mathbf{s}^{\ell}[t]$ is the output of our QSD-Transformer, $\mathcal{L}_{\text{CE}}$ is the cross-entropy loss (Rathi & Roy, 2021) for ensuring task performance, and $\mathcal{L}_{\text{FGD}}$ is the proposed distillation loss for enhancing information entropy. $\lambda$ is a coefficient to balance these two loss functions, and it is set to 2 in this paper.

## 5 EXPERIMENT

In this section, we validate the QSD-Transformer on various vision tasks, including image classification, object detection, semantic segmentation, and transfer learning. Then, we ablate the proposed scheme to prove the effectiveness of our method. For further detailed information on datasets, power calculations, experimental setups, and hyperparameters, refer to Appendix E and G.

**ImageNet classification.** We evaluate the QSD-Transformer's effectiveness in image classification using the challenging ImageNet-1K dataset (Deng et al., 2009). The comparison results are summarized in Table 1. Notably, with only 6.8M parameters, the QSD-Transformer achieves a top-1 accuracy of 80.3% in the SNN domain, showcasing significant advantages in both accuracy and efficiency. Specifically, **QSD-Transformer** vs. SD-Transformer v2 (Yao et al., 2023a) vs. SpikingResformer (Shi et al., 2024): Param, **6.8M** vs. 55.4M vs. 60.4M; Power: **8.7mJ** vs. 52.4mJ vs. 9.7mJ; Acc, **80.3%** vs. 79.7% vs. 78.7%. When compared to the SOTA model in the SNN field, SD-Transformer v2 (Yao et al., 2023a), our method achieves a 0.6% improvement in accuracy while reducing parameter by 87.58% and power by 83.40%. In summary, the QSD-Transformer

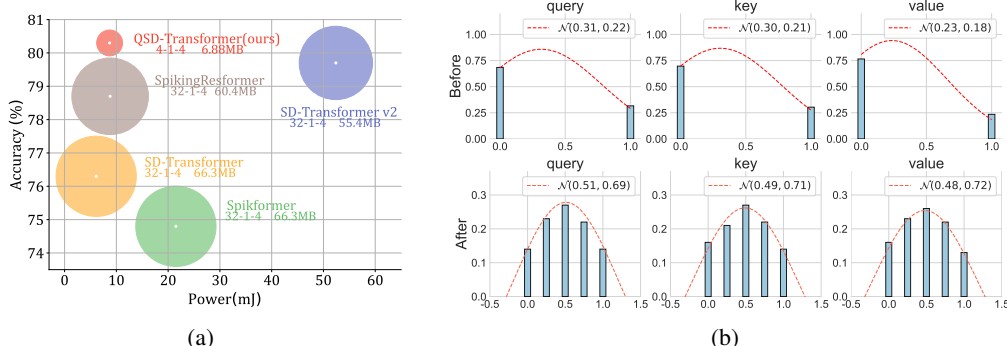

(a)                                    (b)

Figure 3: (a) Comparative results of accuracy, power, and parameters on ImageNet. (b) Comparison of information distribution in Q-SDSA before and after using the proposed IE-LIF and FGD scheme.

Table 1: ImageNet classification results (Deng et al., 2009). ' Bits' denotes the bit width of the weight and activity, respectively. Power is the estimation of energy consumption same as Yao et al. (2023b). $^\star$ indicates self-implementation results with open-source code (Yu et al., 2023).

| Method | Architecture | Bits | Spike -driven | Time Step | Param (M) | Power (mJ) | Acc. (%) |
|---|---|---|---|---|---|---|---|
| Transformer (Yu et al., 2023) | CAformer$^\star$ | 32-32 | ✗ | N/A | 15.1 | 40.3 | 79.9 |
| QCFS (Bu et al., 2021) | ResNet-34 | 32-1 | ✓ | 256 | 21.8 | - | 73.4 |
| MST (Wang et al., 2023) | Swin-T | 32-1 | ✓ | 128 | 28.5 | - | 77.9 |
| SEW-ResNet (Fang et al., 2021) | SEW-ResNet-34 | 32-1 | ✗ | 4 | 25.6 | 4.9 | 67.8 |
|  | SEW-ResNet-152 | 32-1 | ✗ | 4 | 60.2 | 12.9 | 69.2 |
| MS-ResNet(Hu et al., 2024b) | MS-ResNet-34 | 32-1 | ✓ | 4 | 21.8 | 5.1 | 69.4 |
|  | MS-ResNet-104 | 32-1 | ✓ | 4 | 77.3 | 10.2 | 75.3 |
| Spikformer (Zhou et al., 2023b) | Spikformer-8-512 | 32-1 | ✗ | 4 | 29.7 | 11.6 | 73.4 |
|  | Spikformer-8-768 | 32-1 | ✗ | 4 | 66.3 | 21.5 | 74.8 |
| SD-Transformer (Yao et al., 2023b) | SD-Transformer-8-512 | 32-1 | ✓ | 4 | 29.7 | 4.5 | 74.6 |
|  | SD-Transformer-8-768 | 32-1 | ✓ | 4 | 66.3 | 6.1 | 76.3 |
| SpikingResformer (Shi et al., 2024) | SpikingResformer-T | 32-1 | ✓ | 4 | 11.1 | 4.2 | 74.3 |
|  | SpikingResformer-L | 32-1 | ✓ | 4 | 60.4 | 9.7 | 78.7 |
| SD-Transformer v2 (Yao et al., 2023a) | SD-Transformer v2-T | 32-1 | ✓ | 4 | 15.1 | 16.7 | 74.1 |
|  | SD-Transformer v2-M | 32-1 | ✓ | 4 | 31.3 | 32.8 | 77.2 |
|  | SD-Transformer v2-L | 32-1 | ✓ | 4 | 55.4 | 52.4 | 79.7 |
| **QSD-Transformer** | SD-Transformer v2-T | 4-1 | ✓ | 4 | 1.8 | **2.5** | 77.5 |
|  | SD-Transformer v2-M | 4-1 | ✓ | 4 | 3.9 | 5.7 | 78.9 |
|  | SD-Transformer v2-L | 4-1 | ✓ | 4 | 6.8 | 8.7 | **80.3** |

Table 2: Object detection results on COCO 2017 (Lin et al., 2014).

| Method | Architecture | Bits | Spike -driven | Time Step | Param (M) | Power (mJ) | mAP@0.5 (%) |
|---|---|---|---|---|---|---|---|
| Transformer (Yu et al., 2023) | CAformer | 32-32 | ✗ | N/A | 31.2 | 890.6 | 54.0 |
| Transformer (Zhu et al., 2020) | DETR | 32-32 | ✗ | N/A | 41.0 | 860.2 | 57.0 |
| Spiking-Yolo (Kim et al., 2020) | ResNet-18 | 32-1 | ✓ | 3500 | 10.2 | - | 25.7 |
| Spike Calibration (Li et al., 2022) | ResNet-18 | 32-1 | ✓ | 512 | 17.1 | - | 45.3 |
| EMS-SNN (Su et al., 2023) | EMS-ResNet-18 | 32-1 | ✓ | 4 | 26.9 | - | 50.1 |
| SD-Transformer v2 (Yao et al., 2023a) | SD-Transformer v2-M | 32-1 | ✓ | 1 | 75.0 | 140.8 | 51.2 |
| **QSD-Transformer** | SD-Transformer v2-T | 4-1 | ✓ | 4 | 16.9 | **45.1** | 48.1 |
|  | SD-Transformer v2-M | 4-1 | ✓ | 4 | 34.9 | 117.2 | **57.0** |

establishes better results in both accuracy as well as efficiency on ImageNet-1K in the SNN domain and especially shines in efficiency.

**Object detection.** We evaluate the efficacy of the QSD-Transformer on the object detection task and select the classic and large-scale COCO (Lin et al., 2014) dataset as our benchmark for evaluation. Similar to the previous work (Yao et al., 2023a), we convert the *mmdetection* (Chen et al., 2019) codebase into a spiking version by IE-LIF and then use it to execute our model. We employ the

Table 3: Semantic segmentation results on ADE20K (Zhou et al., 2017).

| Method | Architecture | Bits | Spike -driven | Time Step | Param (M) | Power (mJ) | MIoU (%) |
|---|---|---|---|---|---|---|---|
| Segformer (Xie et al., 2021) | Segformer | 32-32 | ✗ | N/A | 3.8 | 38.9 | 37.4 |
| DeepLab-V3 (Zhang et al., 2022a) | DeepLab-V3 | 32-32 | ✗ | N/A | 68.1 | 1240.6 | 42.7 |
| SD-Transformer v2 (Yao et al., 2023a) | SD-Transformer v2-M | 32-1 | ✓ | 4 | 59.8 | 183.6 | 35.3 |
| **QSD-Transformer** | SD-Transformer v2-T | 4-1 | ✓ | 4 | 3.3 | **17.5** | 39.0 |
| | SD-Transformer v2-M | 4-1 | ✓ | 4 | 9.6 | 37.9 | **40.5** |

Table 4: Transfer learning results on CIFAR10, CIFAR100 and CIFAR10-DVS.

| Method | Param (M) | CIFAR10 | | CIFAR100 | | CIFAR10-DVS | |
|---|---|---|---|---|---|---|---|
| | | $T$ | Acc. (%) | $T$ | Acc. (%) | $T$ | Acc. (%) |
| Spikformer (Zhou et al., 2023b) | 29.1 | 4 | 97.0 | 4 | 83.8 | - | - |
| SpikingResformer (Shi et al., 2024) | 17.3 | 4 | 97.4 | 4 | 85.9 | 10 | 84.8 |
| **QSD-Transformer** | 1.8 | 4 | 97.8±0.1 | 4 | 86.6±0.3 | 10 | 88.8±0.1 |
| | 6.8 | 4 | **98.4**±0.2 | 4 | **87.6**±0.2 | 10 | **89.8**±0.1 |

QSD-Transformer as the backbone to extract features, along with Mask R-CNN (He et al., 2017) for object detection. The backbone is initialized with the pre-trained QSD-Transformer on ImageNet-1K, and other added layers are initialized with Xavier (Glorot & Bengio, 2010). The comparison results are summarized in Table 2. Obviously, the QSD-Transformer outperforms the existing state-of-the-art methods in the SNN domain by a significant margin. More specifically, our method exceeds the performance of SD-Transformer v2 by 5.8% in terms of the mAP@0.5 metric, while utilizing fewer than half the parameters. In conclusion, our approach demonstrates efficacy in object detection tasks and has established a new benchmark for detection within the SNN domain.

**Semantic segmentation.** We validate the efficacy of the QSD-Transformer on the semantic segmentation task and select the challenging ADE20K (Zhou et al., 2017) dataset. Similar to the procedures in object detection, we converted the *mmsegmentation* (Contributors, 2020) codebase into a spiking version and utilized it to execute our model. The QSD-Transformer serves as the backbone for feature extraction, integrated with Semantic FPN (Kirillov et al., 2019) for segmentation. The initialization is similar to that in the object detection task. The backbone is initialized with a pre-trained model on ImageNet-1K, and the added layers are initialized using Xavier (Glorot & Bengio, 2010). Since SD-Transformer v2 is the only work in the SNN field reporting results on ADE20K, we compare our approach with advanced deep models. As depicted in Table 3, our method significantly outperforms SD-Transformer v2 (Yao et al., 2023a) across all comparison metrics, achieving an 83.94% reduction in parameters, a 79.36% decrease in power, and a 5.2% increase in MIoU. Moreover, our method achieves a comparable MIoU to the advanced DeepLab-V3 in the ANN domain while substantially reducing both parameters and power.

**Transfer learning.** We demonstrate the efficacy of the QSD-Transformer on transfer learning tasks. We evaluate the model's transfer learning capability on both static datasets (CIFAR) (Krizhevsky et al., 2009) and the neuromorphic dataset (CIFAR10-DVS) (Li et al., 2017) using five repeated experiments with different random seeds. To assess this ability, we fine-tune the pre-trained weights from the ImageNet-1K dataset on these selected datasets. Compared with existing transfer learning methods in SNNs, such as SpikingResformer and Spikformer, the proposed QSD-Transformer demonstrates state-of-the-art results. It achieves 98.4% accuracy on CIFAR-10, 87.6% accuracy on CIFAR-100, and 89.8% accuracy on CIFAR10-DVS, surpassing SpikingResformer by 1.0%, 1.7%, and 5.0%, respectively. Thus, our method achieves the best performance across various computer vision tasks.

**Ablation study.** We first ablate two components of the QSD-Transformer, namely the IE-LIF and FGD schemes, to verify the effectiveness of the proposed method. Additionally, we quantized the weights to 4, 3, and 2 bits to study the impact of bit width on performance. Experiments are performed on the ImageNet dataset. The results are shown in Table 5, where the QSD-Transformer baseline without the IE-LIF neuron and FGD scheme achieves an accuracy of 70.0%. In contrast, using the IE-LIF neuron increases the accuracy by 5.8%. With both the IE-LIF neuron and FGD scheme, the accuracy further reaches 77.5%. Therefore, both the proposed IE-LIF neuron and the FGD scheme

Table 5: Ablation study of the IE-LIF, FGD, and different bits.

| Architecture | IE-LIF | FGD | Weight Bits | Acc.(%) |
|---|---|---|---|---|
| | - | - | 4 | 70.0 |
| | ✓ | - | 4 | 75.8 |
| SD-Transformer v2 (Yao et al., 2023a) | ✓ | ✓ | 4 | **77.5** |
| | ✓ | ✓ | 3 | 76.9 |
| | ✓ | ✓ | 2 | 75.0 |
| | - | - | 4 | 64.1 |
| | ✓ | - | 4 | 70.1 |
| Spikformer (Zhou et al., 2023b) | ✓ | ✓ | 4 | **75.5** |
| | ✓ | ✓ | 3 | 74.1 |
| | ✓ | ✓ | 2 | 73.1 |

can improve performance, and their combined use can bring more significant accuracy. Moreover, we also investigate the impact of bit-width on performance. It can be seen from Table 5 that the accuracy decreases with bit width reduction. Notably, even when the weights are quantized to 2-bit, our method still achieves 75.0% accuracy.

Next, we delve into the application of our method within the Spikformer architecture (Zhou et al., 2023b) to validate its robustness and scalability. Specifically, we initially established a Spikformer-8-384 model as a quantization baseline under the conditions of a time step of 4 and a 4-bit quantization of the weights. Subsequently, we conducted ablation experiments of various modules and weight bit-widths on the ImageNet dataset. As shown in Table 5, direct quantization of Spikformer indicates a significant performance drop of 6.14% under standard quantization conditions. Then, by applying our IE-LIF spiking neurons, we were able to enhance the accuracy by 6.0%. Furthermore, the accuracy was further improved to 75.5% by combining the IE-LIF neurons with the FGD scheme. We also investigated the impact of bit-width on performance. Notably, our method maintains 73.1% accuracy even with 2-bit weight quantization. The above results demonstrate that our method can be robustly applied to various spiking-based Transformer models.

Finally, our ablation studies on the activity bit $b$ and training time step $T$ of the IE-LIF model reveal that augmenting the activity bit $b$ substantially boosts performance. As depicted in Table 6, elevating $b$ from 1 to 4 with $T = 1$ results in a 9.9% increase in accuracy; conversely, with $b = 1$, raising $T$ from 1 to 4 yields a more modest 2.4% improvement. This disparity arises because augmenting the activity bits $b$ enhances the model's information capacity and mitigates quantization errors, whereas increasing the training time step $T$ has a less pronounced impact, likely due to the redundancy inherent in spike trains.

Table 6: Ablation study of the activity bits and training time step on the QSD-Transformer.

| Bits ($b$) | Timestep ($T$) | Acc.(%) |
|---|---|---|
| 1 | 1 | 67.6 |
| 1 | 2 | 68.5 |
| 1 | 4 | 70.0 |
| 2 | 1 | 71.6 |
| 2 | 2 | 77.4 |
| 4 | 1 | **77.5** |

Furthermore, extending the time step $T$ incurs significant memory and energy costs, which is not the case for increasing the activity bit $b$. These findings underscore that the quantization performance is more sensitive to the activity bits than the time step configuration.

## 6 CONCLUSION

In this paper, we first introduce the lightweight spike-driven transformer, namely the QSD-Transformer, which quantifies the weights from 32-bit to low-bit. By employing both low-bit weights and 1-bit spike activities, QSD-Transformer has demonstrated significant energy efficiency. Despite exhibiting efficiency benefits, the QSD-Transformer suffers from performance degradation. We reveal that this is attributed to the SID problem and propose a bi-level optimization strategy to solve this challenge. At the lower level, we propose the IE-LIF neuron, which generates multi-bit spikes in training while maintaining spike-driven behavior during inference. At the upper level, we introduce the FGD scheme, which optimizes attention distribution between the Q-SDSA and its ANN counterpart. Extensive experiments show that our method achieves state-of-the-art results in both

performance and efficiency on various vision tasks, paving the way for the practical deployment of spike-based Transformers in resource-limited platforms.

## ACKNOWLEDGEMENT

This work was supported in part by the National Natural Science Foundation of China under grant U20B2063, 62220106008, and 62106038, the Sichuan Science and Technology Program under Grant 2024NSFTD0034 and 2023YFG0259. The research is supported by Shenzhen Science and Technology Program ZDSYS20230626091302006; and Shenzhen Science and Technology Research Fund (Fundamental Research Key Project Grant No. JCYJ20220818103001002.

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

# Appendix

## A  BACKPROPAGATION PROCESS OF SPIKING NEURONS

There exist two primary methods of training high-performance SNNs. One way is to discretize ANN into spike form through neuron equivalence (Li et al., 2021; Bu et al., 2022; Hao et al., 2023a; Ding et al., 2021), i.e., ANN-to-SNN conversion, but this requires a long simulation time step and boosts the energy consumption. We employ the direct training method (Wu et al., 2018; Qiu et al., 2024b; Wei et al., 2024) and apply surrogate gradient training.

Then in this section, we introduce the training process of SNN gradient descent and the parameter update method of spatio-temporal backpropagation (STBP) (Wu et al., 2018; Xiao et al., 2022). SNNs' parameters can be taught using gradient descent techniques, just like ANNs, after determining the derivative of the generation process. Moreover, the accumulated gradients of loss $\mathcal{L}$ with respect to weights $\mathbf{w}$ at layer $\ell$ can be calculated as:

$$\frac{\partial \mathcal{L}}{\partial \mathbf{w}^\ell} = \sum_{t=1}^{T} \frac{\partial \mathcal{L}}{\partial \mathbf{s}^{\ell+1}[t]} \frac{\partial \mathbf{s}^{\ell+1}[t]}{\partial \mathbf{v}^{\ell+1}[t]} \left( \frac{\partial \mathbf{v}^{\ell+1}[t]}{\partial \mathbf{w}^\ell} + \sum_{\tau < t} \prod_{i=t-1}^{\tau} \left( \frac{\partial \mathbf{v}^{\ell+1}[i+1]}{\partial \mathbf{v}^{\ell+1}[i]} + \frac{\partial \mathbf{v}^{\ell+1}[i+1]}{\partial \mathbf{s}^{\ell+1}[i]} \frac{\partial \mathbf{s}^{\ell+1}[i]}{\partial \mathbf{v}^{\ell+1}[i]} \right) \frac{\partial \mathbf{v}^{\ell+1}[\tau]}{\partial \mathbf{w}^\ell} \right),$$
(15)

where $\mathbf{s}^\ell[t]$ and $\mathbf{v}^\ell[t]$ represent the binary and membrane potential of the neuron in layer $\ell$, at time $t$. Moreover, notice that $\frac{\partial \mathbf{s}^\ell[t]}{\partial \mathbf{v}^\ell[t]}$ is non-differentiable. To overcome this problem, (Wu et al., 2018) propose the surrogate function to make only the neurons whose membrane potentials close to the firing threshold receive nonzero gradients during backpropagation. In this paper, we use the rectangle function, which has been shown to be effective in gradient descent and may be calculated by:

$$\frac{\partial \mathbf{s}^\ell[t]}{\partial \mathbf{v}^\ell[t]} = \frac{1}{a} \operatorname{sign}\left( \left| \mathbf{v}^\ell[t] - \vartheta \right| < \frac{a}{2} \right),$$
(16)

where $a$ is a defined coefficient for controlling the width of the gradient window.

## B  PROOF OF THE PROPOSITION 1.

**Proposition 1.** *Given the SNN Transformer and ANN Transformer models, where the distributions of the query ($\mathbf{q}$), key ($\mathbf{k}$), and value ($\mathbf{v}$) follow binomial $\mathcal{B}(r, T)$ and normal $\mathcal{N}(\mu, \sigma)$ distributions, respectively, it is postulated that as the SNN's time step $T$ tends to infinity, there exist parameters $\mu$, $\sigma$, and $r$ such that the average entropy over time of the SNN's attention scores $\mathcal{H}(\sum_{t=1}^{T} \mathbf{p}^\mathcal{S}[t])$ equals ANN attention scores' entropy $\mathcal{H}(\mathbf{p}^\mathcal{A})$.*

*Proof.* **Proposition 1.** can be restated as follows:

$$\lim_{T \to \infty} \exists \mu, \sigma, r \quad \mathcal{H}(\sum_{t=1}^{T} \mathbf{p}^\mathcal{S}[t]) = \mathcal{H}(\mathbf{p}^\mathcal{A}),$$
(17)

where $\mathbf{p}^\mathcal{A}$ and $\mathbf{p}^\mathcal{S}$ represent the query $\mathbf{q}$, key $\mathbf{k}$, and value $\mathbf{v}$ in the same architecture ANN (teacher) and QSD-Transformer (student), and following the binomial $\mathcal{B}(r, T)$ and normal $\mathcal{N}(\mu, \sigma)$ distributions. $\theta^\mathcal{S}$ is the parameters of the student (QSD-Transformer).

Assume $\mathbf{p}^\mathcal{S}[1], \mathbf{p}^\mathcal{S}[2], \mathbf{p}^\mathcal{S}[3], \ldots, \mathbf{p}^\mathcal{S}[t]$ are $t$ independent random variables, each following a binomial distribution. The expectation is $\mathbb{E}(\mathbf{p}^\mathcal{S}[t]) = r^\mathcal{S}[t]$, where $r^\mathcal{S}[t]$ is the firing rate of the SNN at time $t$. The variance is given by $\mathbb{D}(\mathbf{p}^\mathcal{S}[t]) = \sigma^\mathcal{S}[t]$. And Let $\mathbf{y}^\mathcal{S}[t] = \mathbf{p}^\mathcal{S}[t] - r^\mathcal{S}[t]$, where $\mathbb{E}(\mathbf{y}^\mathcal{S}[t]) = 0$ and $\mathbb{D}(\mathbf{y}^\mathcal{S}[t]) = \sigma$. Let the characteristic function (Chow & Teicher, 2012) of the random variable $\mathbf{y}^\mathcal{S}[t]$ be $\varphi_{\mathbf{y}^\mathcal{S}[t]}(j)$. Then let the random variable $\eta = \frac{\mathbf{y}^\mathcal{S}[1] + \mathbf{y}^\mathcal{S}[2] + \mathbf{y}^\mathcal{S}[3] + \ldots + \mathbf{y}^\mathcal{S}[T]}{\sqrt{t}\sigma}$. Then the characteristic function of $\eta$ is:

$$\varphi_\eta = \left[ \varphi_{\mathbf{y}^\mathcal{S}[t]}(\frac{j}{\sqrt{T}\sigma}) \right] \cdot \left[ \varphi_{\mathbf{y}^\mathcal{S}[t]}(\frac{j}{\sqrt{T}\sigma}) \right] \cdots \left[ \varphi_{\mathbf{y}^\mathcal{S}[t]}(\frac{j}{\sqrt{T}\sigma}) \right] = \left[ \varphi_{\mathbf{y}^\mathcal{S}[t]}(\frac{j}{\sqrt{T}\sigma}) \right]^T,$$
(18)

Then when SNN's timestep $T \to \infty$, $\frac{j}{\sqrt{T}\sigma}$ can be expanded at the point 0 using the Taylor series:

$$\varphi_{\mathbf{y}^{\mathcal{S}}[t]}(\frac{j}{\sqrt{T\sigma}})) = \varphi_{\mathbf{y}^{\mathcal{S}}[t]}(0) + \varphi'_{\mathbf{y}^{\mathcal{S}}[t]}(0)\left(\frac{j}{\sqrt{T\sigma}}\right) + \frac{\varphi''_{\mathbf{y}^{\mathcal{S}}[t]}(0)}{2!}\left(\frac{j}{\sqrt{T\sigma}}\right)^2 + o\left(\left(\frac{j}{\sqrt{T\sigma}}\right)^2\right),$$

Since $\varphi_{\mathbf{y}^{\mathcal{S}}[t]}(0) = 1$, $\varphi'_{\mathbf{y}^{\mathcal{S}}[t]}(0) = 0$, and $\varphi''_{\mathbf{y}^{\mathcal{S}}[t]}(0) = -\sigma$, we have:

$$\varphi_{\mathbf{y}^{\mathcal{S}}[t]}(\frac{j}{\sqrt{T\sigma}}) = 1 - \frac{j^2}{2T} + o\left(\left(\frac{j}{\sqrt{T\sigma}}\right)^2\right),$$

$$\left[\varphi_{\mathbf{y}^{\mathcal{S}}[t]}(\frac{j}{\sqrt{T\sigma}})\right]^T = \left[1 - \frac{j^2}{2T} + o\left(\left(\frac{j}{\sqrt{T\sigma}}\right)^2\right)\right]^T = \left[1 - \frac{j^2}{2T} + o\left(\left(\frac{j}{\sqrt{T\sigma}}\right)^2\right)\right],$$

Hence:

$$\lim_{T\to\infty}\left[\varphi_{\mathbf{y}^{\mathcal{S}}[t]}(\frac{j}{\sqrt{T\sigma}})\right]^T = \lim_{T\to\infty}\left[1 - \frac{j^2}{2T} + o((\frac{j}{\sqrt{T\sigma}}))\right]^T = e^{-\frac{j^2}{2}}, \tag{19}$$

where $e^{-\frac{j^2}{2}}$ happens to be the characteristic function of a random variable following the standard normal distribution $\mathcal{N}(0,1)$, so $\eta$ follows the standard normal distribution, which distribution is the same to the attention score in ANN Transformer. Hence, as the SNN's time step $T$ tends to infinity, $\exists \mu, \sigma, r$ such that $\mathcal{H}(\sum_{t=1}^{T}\mathbf{p}^{\mathcal{S}}[t]) = \mathcal{H}(\mathbf{p}^{\mathcal{A}})$. $\square$

## C  PROOF OF THE PROPOSITION 2.

**Proposition 2.** *For a random variable* $\mathbf{x} \sim \mathcal{N}(\mu, \sigma)$, *the information entropy* $\mathbf{x}$ *reaches its maximum value* $\mathcal{H}(\mathbf{x}) = \frac{1}{2}\log 2\pi e\sigma^2(\mathbf{x})$ *and is observed to increase with the expansion of variance* $\sigma$.

*Proof.* For a continuous random variable $\mathbf{x}$ obeying a normal distribution, its probability density function $p(x)$ is given by:

$$p(x) = \frac{1}{(2\pi\sigma^2)^{1/2}}\exp\left\{-\frac{(x-\mu)^2}{2\sigma^2}\right\}, \tag{20}$$

Consequently, the differential entropy of $\mathbf{x}$ can be calculated as

$$\mathcal{H}(\mathbf{x}) = -\int_{-\infty}^{\infty} p(x)\log p(x)dx,$$

$$= -\int \frac{1}{(2\pi\sigma^2)^{1/2}}\exp\left\{-\frac{(x-\mu)^2}{2\sigma^2}\right\}\log\frac{1}{(2\pi\sigma^2)^{1/2}}\exp\left\{-\frac{(x-\mu)^2}{2\sigma^2}\right\}dx,$$

$$= -\frac{1}{(2\pi\sigma^2)^{1/2}}\int \exp\left\{-\frac{(x-\mu)^2}{2\sigma^2}\right\}\left(-\log\left(\sqrt{2\pi}\sigma\right) - \frac{(x-\mu)^2}{2\sigma^2}\right)dx,$$

$$= -\frac{1}{(2\pi\sigma^2)^{1/2}} \cdot -\log\left(\sqrt{2\pi}\sigma\right)\int \exp\left\{-\frac{(x-\mu)^2}{2\sigma^2}\right\}dx +$$

$$\frac{1}{(2\pi\sigma^2)^{1/2}}\int \exp\left\{-\frac{(x-\mu)^2}{2\sigma^2}\right\}\frac{(x-\mu)^2}{2\sigma^2}dx,$$

$$= \frac{\log\left(\sqrt{2\pi}\sigma\right)}{(2\pi\sigma^2)^{1/2}}\int \exp\left\{-\frac{(x-\mu)^2}{2\sigma^2}\right\}dx + \frac{1}{(2\pi\sigma^2)^{1/2}}\int \exp\left\{-\frac{(x-\mu)^2}{2\sigma^2}\right\}\frac{(x-\mu)^2}{2\sigma^2}dx,$$

$$= \frac{\log\left(\sqrt{2\pi}\sigma\right)}{(2\pi\sigma^2)^{1/2}}\sqrt{2}\sigma\int \exp\left\{-\left(\frac{x-\mu}{\sqrt{2}\sigma}\right)^2\right\}d\left(\frac{x-\mu}{\sqrt{2}\sigma}\right) +$$

$$\frac{1}{(2\pi\sigma^2)^{1/2}}\sqrt{2}\sigma\int \exp\left\{-\left(\frac{x-\mu}{\sqrt{2}\sigma}\right)^2\right\}\frac{(x-\mu)^2}{2\sigma^2}d\left(\frac{x-\mu}{\sqrt{2}\sigma}\right). \tag{21}$$

Moreover, it can be easily proven that

$$\int_{-\infty}^{\infty} e^{-y^2} dy = \sqrt{\pi}. \tag{22}$$

Thus,

$$
\begin{aligned}
\mathcal{H}(\mathbf{x}) &= \frac{\log\left(\sqrt{2\pi}\sigma\right)}{\sqrt{\pi}} \int_{-\infty}^{\infty} e^{-y^2} dy + \frac{1}{\sqrt{\pi}} \int_{-\infty}^{\infty} e^{-y^2} y^2 dy, \\
&= \log\left(\sqrt{2\pi}\sigma\right) + \frac{1}{\sqrt{\pi}} \cdot -\frac{1}{2}\left(0 - \int_{-\infty}^{\infty} e^{-y^2} dy\right), \\
&= \log\left(\sqrt{2\pi}\sigma\right) + \frac{1}{2}, \\
&= \frac{1}{2}\Big(\log\left(2\pi\sigma^2\right) + 1\Big), \\
&= \frac{1}{2}\log\left(2\pi e\sigma^2\right).
\end{aligned}
\tag{23}
$$

$\square$

## D  THEORETICAL ANALYSIS IN FUSION OF MPRF AND WEIGHTS

In Section 4.2, we introduce a membrane potential rectify function (MPRF) $\phi^\ell(\cdot)$ aimed at maximizing the information entropy of the attention score. The inherent homogeneity of convolution operations permits the subsequent batch normalization (BN) and linear scaling transformations to be seamlessly integrated into the convolutional layer with an added bias during deployment. This approach enables the model to conduct inference rapidly without incurring additional computational overhead. Specifically, we utilize Eq. 24 to represent the quantized convolution (Q-Conv):

$$\mathbf{y}_Q^\ell = \mathbf{w}_{Q_{Conv}}^\ell \cdot \mathbf{S}^\ell + \mathbf{b}_{Q_{Conv}}^\ell \tag{24}$$

where $\mathbf{S}$ denotes input binary spike, $\mathbf{w}_{Q_{Conv}}$ and $\mathbf{b}_{Q_{Conv}}$ are quantized weights and bias of the Q-Conv layer. $\mathbf{y}_Q$ denotes the output of the Q-Conv layer. After employing MPRF, the rectified output should be computed as Eq. 25

$$
\begin{aligned}
\hat{\mathbf{y}}_Q^\ell &= \phi(\mathbf{y}_Q^\ell) = \phi(\mathbf{w}_{Q_{Conv}}^\ell \cdot \mathbf{S}^\ell + \mathbf{b}_{Q_{Conv}}^\ell), \\
&= \frac{(\mathbf{w}_{Q_{Conv}}^\ell \cdot \mathbf{S}^\ell + \mathbf{b}_{Q_{Conv}}^\ell) - \mu(\mathbf{w}_{Q_{Conv}}^\ell \cdot \mathbf{S}^\ell + \mathbf{b}_{Q_{Conv}}^\ell)}{\sigma(\mathbf{w}_{Q_{Conv}}^\ell \cdot \mathbf{S}^\ell + \mathbf{b}_{Q_{Conv}}^\ell)} \cdot \gamma^\ell + \alpha^\ell, \\
&= \frac{\gamma^\ell \cdot (\mathbf{w}_{\mathbf{Q_{Conv}}}^\ell \cdot \mathbf{S}^\ell + \mathbf{b}_{Q_{Conv}}^\ell)}{\sigma(\mathbf{w}_{Q_{Conv}}^\ell \cdot \mathbf{S}^\ell + \mathbf{b}_{Q_{Conv}}^\ell)} - \frac{\gamma^\ell \cdot \mu(\mathbf{w}_{Q_{Conv}}^\ell \cdot \mathbf{S}^\ell + \mathbf{b}_{Q_{Conv}}^\ell)}{\sigma(\mathbf{w}_{Q_{Conv}}^\ell \cdot \mathbf{S}^\ell + \mathbf{b}_{Q_{Conv}}^\ell)} + \alpha^\ell, \\
&= \frac{\gamma^\ell \cdot \mathbf{w}_{Q_{Conv}}^\ell}{\sigma(\mathbf{w}_{Q_{Conv}}^\ell \cdot \mathbf{S}^\ell + \mathbf{b}_{Q_{Conv}}^\ell)} \cdot \mathbf{S}^\ell + [\frac{\gamma^\ell \cdot \mathbf{b}_{Q_{Conv}}^\ell - \mu(\mathbf{w}_{Q_{Conv}}^\ell \cdot \mathbf{S}^\ell + \mathbf{b}_{Q_{Conv}}^\ell)}{\sigma(\mathbf{w}_{Q_{Conv}}^\ell \cdot \mathbf{S}^\ell + \mathbf{b}_{Q_{Conv}}^\ell)} + \alpha^\ell], \\
&= \mathbf{w}_{\mathbf{f}}^\ell \cdot \mathbf{S} + \mathbf{b}_{\mathbf{f}}^\ell,
\end{aligned}
\tag{25}
$$

where $\mathbf{w}_{\mathbf{f}}^\ell$ and $\mathbf{b}_{\mathbf{f}}^\ell$ denote the fusioned weight and bias:

$$
\begin{aligned}
\mathbf{w}_{\mathbf{f}}^\ell &= \frac{\gamma^\ell \cdot \mathbf{w}_{Q_{Conv}}^\ell}{\sigma(\mathbf{w}_{Q_{Conv}}^\ell \cdot \mathbf{S}^\ell + \mathbf{b}_{Q_{Conv}}^\ell)}, \\
\mathbf{b}_{\mathbf{f}}^\ell &= \frac{\gamma^\ell \cdot \mathbf{b}_{Q_{Conv}}^\ell - \mu(\mathbf{w}_{Q_{Conv}}^\ell \cdot \mathbf{S}^\ell + \mathbf{b}_{Q_{Conv}}^\ell)}{\sigma(\mathbf{w}_{Q_{Conv}}^\ell \cdot \mathbf{S}^\ell + \mathbf{b}_{Q_{Conv}}^\ell)} + \alpha^\ell.
\end{aligned}
\tag{26}
$$

## E  THEORETICAL ENERGY CONSUMPTION ANALYSIS

When disregarding the energy consumption factors related to hardware manufacturing processes, data access, and storage, comparing the computational energy consumption of different models remains

compelling. Such comparisons effectively reflect the intrinsic computational efficiency of various network models. Previous work by (Horowitz, 2014) indicates that, on a 45nm process hardware platform, the energy consumption for a single multiply-accumulate (MAC) operation is 4.6pJ (with 3.7pJ for multiplication and 0.9pJ for addition). Many performance analyses in the research of spiking neural networks (SNNs) (Panda et al., 2020; Qiu et al., 2024a; Shan et al., 2024) also reference this data.

### E.1 COMPARISION ON MHSA AND SDSA

Given a float-point input sequence $X \in \mathbb{R}^{N \times D}$, the float-point Query ($\mathbf{q}$), Key ($\mathbf{k}$), and Value ($\mathbf{v}$) in $\mathbb{R}^{N \times D}$ are computed using three learnable linear matrices, where $N$ is the token number, and $D$ is the channel dimension. The MHSA scaled dot-product is computed as described by (Dosovitskiy et al., 2020):

$$\text{MHSA}(\mathbf{q}, \mathbf{k}, \mathbf{v}) = \mathbf{softmax}\left(\frac{\mathbf{q}\mathbf{k}^{\text{T}}}{\sqrt{d}}\right)\mathbf{v} \tag{27}$$

where $d = \frac{D}{H}$ is the feature dimension of one head and H is the number of heads, and $\sqrt{d}$ serves as the scaling factor. Typically, MHSA divides $\mathbf{q}$, $\mathbf{k}$ and $\mathbf{v}$ into $H$ heads along the channel dimension. For the $i^{th}$ head, $\mathbf{q_i}$, $\mathbf{k_i}$ and $\mathbf{v_i}$ are in $\mathbb{R}^{N \times D/H}$. After performing the self-attention operation on each of the $H$ heads independently, the outputs are concatenated together.

In MHSA, $\mathbf{q}$ and $\mathbf{k}$ are matrix-multiplied, followed by a matrix multiplication of their output with $\mathbf{v}$. The computational complexity of MHSA($\cdot$) is $O(N^2 D)$, indicating a *quadratic* relationship with the token number $N$. For the SDSA modules, the computational complexity is $O(ND^2)$, and the energy cost of the Rep-Conv part is consistent with SNN-based convolution. The energy cost of the SDSA operator part is given in Table 7.

Table 7: Theoretical FLOPs/SOPs of self-attention modules.

| Function | Multi-head Self-attention (MHSA) $\text{MHSA}(\mathbf{q}, \mathbf{k}, \mathbf{v}) = \mathbf{softmax}\left(\frac{\mathbf{q}\mathbf{k}^{\text{T}}}{\sqrt{d}}\right)\mathbf{v}$ | Spike-driven Self-attention (SDSA) $\text{SDSA}(\mathbf{q_s}, \mathbf{k_s}, \mathbf{k_s}) = \mathcal{SN}_s((\mathbf{q_s}\mathbf{k_s}^{\text{T}})\mathbf{v_s})$ |
|---|---|---|
| $q, k, v$ | $3ND^2$ | $T \cdot fr_1 \cdot 3 \cdot FL_{Conv}$ |
| $f(q, k, v)$ | $2N^2 D$ | $T \cdot fr_2 \cdot ND^2$ |
| Scale | $N^2$ | - |
| Softmax | $2N^2$ | - |
| Linear | $FL_{fc}$ | $T \cdot fr_3 \cdot FL_{fc}$ |

### E.2 THEORETICAL ENERGY CONSUMPTION OF QSD-TRANSFORMER

We first calculate the theoretical energy consumption requires calculating the synaptic operations (SOPs):

$$\text{SOPs}^{\ell} = fr^{\ell} \times T \times \text{FLOPs}^{\ell} \tag{28}$$

where $fr^{\ell}$, $\text{FLOPs}^{\ell}$, and $T$ is the firing rate, float-pointing operations, and timestep of layer $\ell$. Moreover, the respective number of FLOPs adds $\{\frac{1}{32}, \frac{1}{16}, \frac{1}{8}\}$ of the number of {2,3,4}-bit multiplications equals the OPs following (Liu et al., 2020; Qin et al., 2020).

The total energy consumption of the network can be calculated using Eq. 29 for non-quantized models and Eq. 30 for quantized models:

$$E_{total} = E_{MAC} \cdot \text{FLOPs}^1_{conv} + E_{AC} \cdot T \cdot \left(\sum_{n=2}^{N} \text{FLOPs}^n_{conv} \cdot fr^n + \sum_{m=1}^{M} \text{FLOPs}^m_{fc} \cdot fr^m\right), \tag{29}$$

$$E_{total} = E_{MAC} \cdot \text{FLOPs}^1_{conv} + E_{AC} \cdot \left(\sum_{n=2}^{N} \text{SOPs}^N + \sum_{m=1}^{M} \text{SOPs}^M\right) \tag{30}$$

where $N$ and $M$ are the total number of Conv and FC layers, $E_{MAC}$ and $E_{AC}$ are the energy costs of MAC and AC operations, and $fr^m$, $fr^n$, $\text{FLOPs}^n_{conv}$ and $\text{FLOPs}^m_{fc}$ are the firing rate and FLOPs

of the $n$-th Conv and $m$-th FC layer. Previous SNN works (Horowitz, 2014; Rathi & Roy, 2021; Hu et al., 2024a) assume 32-bit floating-point implementation in 45nm technology, where $E_{MAC}$ = 4.6pJ and $E_{AC}$ = 0.9pJ for various operations.

## F    LIMITATIONS AND FUTURE WORKS

**Limitations**    The limitations of this work include the scalability of low-bit spike-driven Transformer models and the hardware deployment, which we will address in future research. The experimental results presented in this paper are reproducible. Detailed explanations of model training and configuration are provided in the main text and supplemented in the appendix. Our codes and models will be made available on GitHub after review.

**Future Works**    Since the largest Spikformer-V2 Zhou et al. (2024c) model has not yet released its training code and weights, we will attempt to quantify the Spikformer-V2 model in the future to demonstrate the scalability of our approach. Moreover, we did not take the energy consumption of memory access into account when calculating the theoretical energy consumption owing to the diversity of different dataflow and memory access schemes and the implementation on various hardware platforms. We will deploy our lightweight model into hardware platforms such as Field Programmable Gate Arrays (FPGAs) to evaluate the factual performance, where we will optimize the suitable read-write data streams and memory access schemes to enhance the inference speed of the models.

## G    EXPERIMENT DETAILS

### G.1    IMAGENET-1K EXPERIMENTS

ImageNet-1K dataset is commonly used for computer vision tasks. It spans 1000 object classes and contains around 1.3 million training images and 50,000 validation images. For experiments on the ImageNet dataset, we used the hyper-parameters shown in Table 8. Moreover, we employ our model on three different scales, with the specific model configurations detailed in Table 9. We conducted training on eight 40GB A100 GPUs. For the three different model scales—1.8M, 3.8M and 6.8M parameters—we allocated 24, 28 and 36 hours of training time, respectively.

Table 8: Hyper-parameters for image classification on ImageNet-1K and CIFAR10/100.

| Hyper-parameter | ImageNet | CIFAR10/10 |
| --- | --- | --- |
| Timestep (Training/Inference) | 1/4 | 1/4 |
| Epochs | 300 | 100 |
| Resolution | 224×224 | 128×128 |
| Batch size | 1568 | 256 |
| Optimizer | LAMB | LAMB |
| Base Learning rate | 6e-4 | 6e-4 |
| Learning rate decay | Cosine | Cosine |
| Warmup eopchs | 10 | None |
| Weight decay | 0.05 | 0.05 |
| Rand Augment | 9/0.5 | 9/0.5 |
| Mixup | None | 0.8 |
| Cutmix | None | 1.0 |
| Label smoothing | 0.1 | None |

### G.2    COCO EXPERIMENTS

The COCO dataset aims at scene understanding, primarily extracted from complex everyday scenes, where objects in images are precisely localized through accurate segmentation. The COCO dataset comprises 118K training images and 5K validation images. In the COCO experiments, we pre-trained the QSD-Transformer on ImageNet-1k as the backbone, and then fine-tuned it on the COCO dataset for 24 epochs with the Mask R-CNN as detector to obtain the final model. During the fine-tuning

Table 9: Configurations of different QSD-Transformer models, which is similar to our strong baseline Spike-driven Transformer v2 (Yao et al., 2023a).

| stage | # Tokens | Layer Specification | | | 1.8M | 3.8M | 6.8M |
|---|---|---|---|---|---|---|---|
| 1 | $\frac{H}{2} \times \frac{W}{2}$ | Downsampling | | Conv | 7x7 stride 2 | | |
| | | Downsampling | | Dim | 32 | 48 | 64 |
| | | Conv-based SNN block | SepConv | DWConv | 7x7 stride 1 | | |
| | | | SepConv | FC ratio | 2 | | |
| | | | Channel Conv | Conv | 3x3 stride 1 | | |
| | | | Channel Conv | Conv ratio | 4 | | |
| | $\frac{H}{4} \times \frac{W}{4}$ | Downsampling | | Conv | 3x3 stride 2 | | |
| | | Downsampling | | Dim | 64 | 96 | 128 |
| | | Conv-based SNN block | SepConv | DWConv | 7x7 stride 1 | | |
| | | | SepConv | FC ratio | 2 | | |
| | | | Channel Conv | Conv | 3x3 stride 1 | | |
| | | | Channel Conv | Conv ratio | 4 | | |
| 2 | $\frac{H}{8} \times \frac{W}{8}$ | Downsampling | | Conv | 3x3 stride 2 | | |
| | | Downsampling | | Dim | 128 | 192 | 256 |
| | | Conv-based SNN block | SepConv | DWConv | 7x7 stride 1 | | |
| | | | SepConv | FC ratio | 2 | | |
| | | | Channel Conv | Conv | 3x3 stride 1 | | |
| | | | Channel Conv | Conv ratio | 4 | | |
| | | | # Blocks | | 2 | | |
| 3 | $\frac{H}{16} \times \frac{W}{16}$ | Downsampling | | Conv | 3x3 stride 2 | | |
| | | Downsampling | | Dim | 256 | 384 | 512 |
| | | Transformer-based SNN block | SDSA | RepConv | 3x3 stride 1 | | |
| | | | Channel FC | FC ratio | 4 | | |
| | | | # Blocks | | 6 | | |
| 4 | $\frac{H}{16} \times \frac{W}{16}$ | Downsampling | | Conv | 3x3 stride 1 | | |
| | | Downsampling | | Dim | 360 | 480 | 640 |
| | | Transformer-based SNN block | SDSA | RepConv | 3x3 stride 1 | | |
| | | | Channel FC | FC ratio | 4 | | |
| | | | # Blocks | | 2 | | |

stage, we resized and cropped the training and test data to 1333x800. Additionally, we applied random horizontal flipping and resize with a ratio of 0.5 to the training data. The batch size was set to 12. We used the AdamW optimizer with an initial learning rate of 1e-4, and the learning rate was decayed polynomially with a power of 0.9. We conducted training on four 40GB A100 GPUs for a duration of 26 hours.

### G.3 ADE20K EXPERIMENTS

The ADE20K semantic segmentation dataset comprises over 20K training and 2K validation scene-centric images meticulously annotated with pixel-level object and object parts labels, fostering a comprehensive understanding of complex scenes. It encompasses a total of 150 semantic categories, encompassing elements such as sky, road, and grass, as well as discrete entities like person, car, and bed. We also used the QSD-Transformer pre-trained on ImageNet-1K as the backbone combined with FPN for segmentation experiments. The newly added parameters were initialized using Xavier initialization, and the model was trained on the ADE20K dataset with a batch size of 20 for 160K iterations. We utilized the AdamW optimizer with an initial learning rate of $1 \times 10^{-4}$, and the learning rate was decayed polynomially with a power of 0.9. During the initial 1500 iterations, we employed linear decay to warm up the model. The training process was executed on four 40GB A100 GPUs and lasted for 25 hours.

### G.4 TRANSFER LEARNING

We performed transfer learning experiments on the static image classification datasets CIFAR10/100 and the neuromorphic classification dataset CIFAR10-DVS. The CIFAR10/100 datasets each have

50,000 training and 10,000 test images with a resolution of $32 \times 32$. CIFAR10-DVS consists of 10K event streams created by capturing CIFAR10 images using a DVS camera.

In these experiments, we first loaded pre-trained ImageNet-1K checkpoints and replaced the final fully connected layer to match the number of classes in each dataset (e.g., replacing the 1000-FC with 100-FC for CIFAR-100). During fine-tuning, we applied data augmentations like mixup, cutmix, and label smoothing. We used a batch size of 128, the AdamW optimizer with a weight decay of 0.01, and a cosine-decay learning rate schedule starting at $1 \times 10^{-4}$ over 100 epochs. The experiments were run on a single 32GB V100 GPU, taking 12 hours for CIFAR-10 and CIFAR-100, and 10 hours for CIFAR-10-DVS.

For CIFAR10-DVS, we added preprocessing steps: dividing the event stream into $T$ slices, each with an equal number of events, and compressing these into three-channel frames representing positive, negative, and all events, transforming the event stream into $T$ frames. We also applied data augmentation to the processed event data, as described in (Wang et al., 2023; Shi et al., 2024).

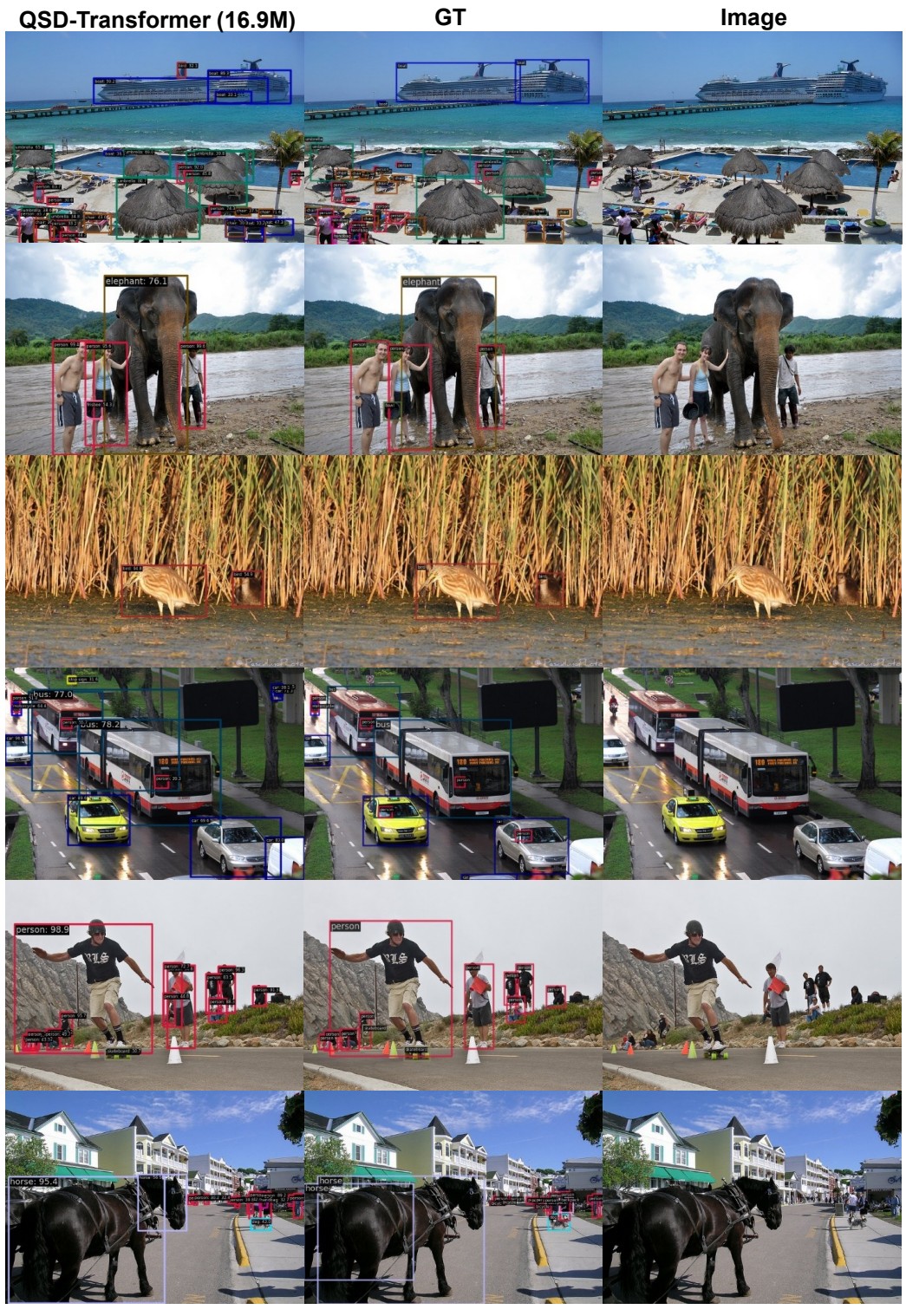

Figure 4: Visualization of results on COCO dataset. Our QSD-Transformer excels in the target detection task.

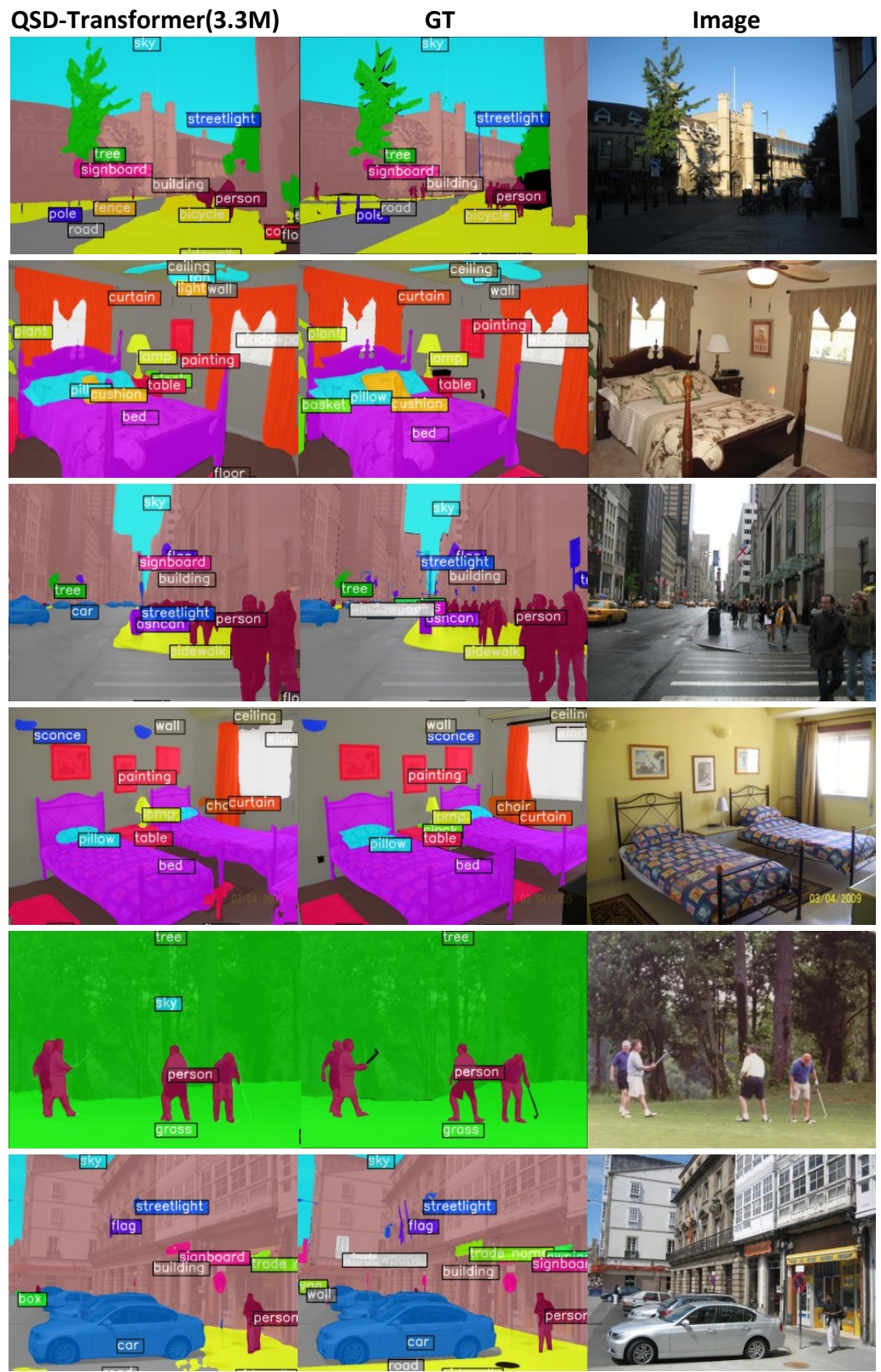

Figure 5: Visualization of results on ADE20K dataset. Our QSD-Transformer excels in the segmentation task.

