# OpenReview forum: "Quantized Spike-driven Transformer"
_ICLR.cc/2025/Conference — ICLR 2025 Poster_

### Official Review · Reviewer_ynnK · 2024-10-28

**Soundness:** 3
**Presentation:** 3
**Contribution:** 3
**Rating:** 5
**Confidence:** 4

**Summary:**

The author proposed Quantized Spike-Driven Transformer(QSD-Transformer) to tackle with the spike information distortion (SID) challenge resulted from quantized spike-driven self-attention (Q-SDSA). The author addressed the problems through two levels: 1) Information-Enhanced LIF(IE-LIF) neuron to rectify the information distribution in Q-SDSA at the lower level. 2) A fine- grained distillation scheme for the QSD-Transformer to align the distribution in Q-SDSA with that in the counterpart ANN at the upper level. QSD-Transformer achieved promising results on multiple computer vision tasks.

**Strengths:**

1. Extensive experiments and ablation studies on Image Classification, Object Detection and Semantic Segmentation.
2. The proposed Information-Enhanced LIF(IE-LIF) neuron is effective to rectify the information distribution in Q-SDSA through Information Theory.
3. Clear writing and methodology.

**Weaknesses:**

1. The comparison between ANN2SNN and Direct Training methods is limited. Currently, MST is no longer the SOTA method on ANN2SNN method. SpikeZIP-TF (ICML 2024) [1] and ECMT (ACM MM 2024) achieve better performance on Image Classification tasks. The performance of  SpikeZIP-TF and ECMT on ImageNet surpass QSD-Transformer by a large margin. In addition, ANN2SNN methods has advantage on saving computational resources compared with Direct Training methods. It is recommended that the author should conduct a more comprehensive comparison between those two methods.
2. The method proposed by the author is somewhat cumbersome, did the training time provided by the authors in appendix include the training time of FGD?
3.  It is recommended that the author should extend the method to NLP tasks to verify the transferability of QSD-Transformer.

[1] Kang You*, Zekai Xu* et al. SpikeZIP-TF: Conversion is All You Need for Transformer-based SNN. International Conference on Machine Learning 2024
[2] Zihan Huang, Xinyu Shi et al. Towards High-performance Spiking Transformers from ANN to SNN Conversion. ACM Multimedia 2024

**Questions:**

1. I wonder whether you quantized the membrane potential or not. If you didn't quantize the membrane potential, it seems hard to implement your method on hardware.

---

### Official Review · Reviewer_Uosk · 2024-11-01

**Soundness:** 3
**Presentation:** 3
**Contribution:** 3
**Rating:** 6
**Confidence:** 4

**Summary:**

This paper proposes a quantized spike-driven transformer and increases the accuracy of the SNN by proposing an information-enhanced LIF model and fine-grained distillation from the lower level and upper level respectively. Experiments show that these technologies reduce energy consumption significantly while increasing the accuracy of SNN.

**Strengths:**

1. The paper is well-written and easy to follow.
2. The IE-LIF neuron combines the conversion algorithm and training algorithm, which is novel.
3. The experimental results are significant.

**Weaknesses:**

1. The training of IE-LIF neurons does not utilize temporal information, which is not suitable for temporal benchmarks.
2. The reason why the IE-LIF neuron and fine-grained distillation that reduces the energy consumption is not provided.

**Questions:**

1. Why do the IE-LIF neuron and fine-grained distillation that reduce the energy consumption? Do these technologies reduce the number of synaptic operations?
2. Why the energy reduction in the COCO2017 dataset is not significant?

---

### Official Review · Reviewer_iFNE · 2024-11-03

**Soundness:** 3
**Presentation:** 3
**Contribution:** 3
**Rating:** 8
**Confidence:** 5

**Summary:**

To tackle the issues of high-parameter spike-based Transformer in resource-constrained applications and the low performance of directly quantized spike-based Transformers, this paper introduces a Quantized Spike-driven Transformer. It uses a bi-level optimization strategy, including an Information-Enhanced LIF neuron and fine-grained distillation, to counteract quantization-induced performance degradation. The comparative and ablation experiments demonstrate the effectiveness of the proposed methods.

**Strengths:**

1. Based on Information entropy, this paper proposes a bi-level optimization strategy, which mitigates quantization-induced performance drops in baseline quantized spiking Transformers.

2. The proposed IE-LIF spike neuron is hardware-friendly and converts to a binary-spiking LIF neuron during inference to maintain spike-driven nature.

3. Experimental results on ImageNet and a large number of vision tasks (detection, segmentation, and transfer learning) show that the method is effective and energy-efficient on various spike-based transformers.

**Weaknesses:**

1. The authors describe the implementation method of surrogate gradients for binary spikes in the appendix. However, the proposed IE-LIF in the main text is multi-bit. Could the authors explain how the aforementioned surrogate gradients are deployed in the proposed neurons?

2. Fast-SNN [1] converts quantized ANNs to SNNs and is a competitive ANN-to-SNN method. Like this paper, it aims to reduce quantization error and employs quantization techniques to enhance energy efficiency. The basic idea of Fast-SNN is to reduce both quantization error and accumulating error, achieving excellent performance on many vision tasks (classification, detection, and segmentation). Could you include comparative experimental results with Fast-SNN?

3. There are some typos in the text, such as the equation (1) on line 164  and on line 292, it seems you intended to reference Fig 1(b).

[1] Hu, Yangfan, et al. "Fast-SNN: fast spiking neural network by converting quantized ANN." IEEE Transactions on Pattern Analysis and Machine Intelligence, vol. 45, no. 12, pp. 14546-14562, Dec. 2023, doi: 10.1109/TPAMI.2023.3275769.

**Questions:**

See Weakness.

**Details Of Ethics Concerns:**

None.

---

> ### Comment · Reviewer_iFNE · 2024-11-19
> **Response to authors' rebuttal**
>
> I appreciate the authors' thorough response, which has addressed some of my concerns. However, I still have further questions and comments.
> 1. Can the training efficiency be improved with this approach when compared with the original spiking transformer?
> 2. Is it feasible for this method to be effectively applied to NLP tasks?

---

> ### Comment · Reviewer_iFNE · 2024-11-21
>
> Many thanks to the author's rebuttal for solving my problem. So I'm willing to raise my score.

---

### Official Review · Reviewer_pkTr · 2024-11-03

**Soundness:** 3
**Presentation:** 3
**Contribution:** 2
**Rating:** 6
**Confidence:** 5

**Summary:**

This paper presents QSD-Transformer, a quantized spike-driven transformer that addresses the challenge of implementing efficient spiking neural networks (SNNs) while maintaining performance. The work shows three key points: (1) a lightweight quantized baseline for spike-driven transformers, (2) a bi-level optimization strategy to address spike information distortion (SID), and (3) information-enhanced LIF (IE-LIF) neurons with fine-grained distillation for improved performance.

**Strengths:**

- First approach to quantizing spike-based transformers with SID analysis
- Solid theoretical analysis and proofs for the proposed methods
- Competitive accuracy on ImageNet (80.3%) with low energy
- Extensive experiments on multiple tasks (classification, object detection, segmentation)

**Weaknesses:**

- Lack of comparison with the previous SNN and quantized ANN transformer models (Connected to question #1)
- Limited scalability in ImageNet experiments (Connected to question #2)
- Huge training overhead compared to the conventional spike-based transformer due to multi-bit spike and knowledge distillation (Connected to question #3)

**Questions:**

1. To make the ImageNet results table solid, authors can add additional results such as QKFormer [1] and previous QAT-ViT models [2].
2. I just wondered why only small sizes of architecture (1.8M, 3.9M, 6.8M) are used for training the ImageNet dataset. Is there any scalability issue with this method?
3. This work uses multi-bit spikes during training and knowledge distillation with ANN architecture, which causes huge training overhead in training time and memory. Can you present any analysis of this overhead?
4. In the transfer learning section, the overall information is insufficient. Which bit-width did you use? and could you provide us the accuracy of CIFAR10/100, and CIFAR10-DVS without transfer learning?
5. Can the authors provide the firing rate information? Compared to the original Spike-driven Transformer-V2, how has the firing rate changed in the self-attention part?

[1] Zhou, Chenlin, et al. "QKFormer: Hierarchical Spiking Transformer using QK Attention." arXiv preprint arXiv:2403.16552 (2024).
[2] Li, Yanjing, et al. "Q-vit: Accurate and fully quantized low-bit vision transformer." Advances in neural information processing systems 35 (2022): 34451-34463.

---

### Official Review · Reviewer_xmWz · 2024-11-13

**Soundness:** 3
**Presentation:** 2
**Contribution:** 2
**Rating:** 5
**Confidence:** 5

**Summary:**

This paper presents a Quantized Spike-Driven Transformer, addressing the spike information distortion during quantization caused by the bimodal distribution of Quantized Spike-Driven Self-Attention (Q-SDSA). A bi-level optimization strategy is introduced, incorporating information-enhanced LIF and fine-grained distillation to rectify the distribution of Q-SDSA.

**Strengths:**

1. This paper aims to address the problem of performance degradation of the Spike Transformer after quantization, attributing it to the spiking information distortion (SID) problem. The authors presents a loss function based on mutual information maximization to tackle the problem.
2. The authors conduct experiments across various tasks to demonstrate the effectiveness of the proposed method.
3. The paper is well organized and easy to follow.

**Weaknesses:**

1. The primary reason for the improved quantization performance of the proposed method is the use of multi-bit spikes instead of traditional 0-1 spikes. Specifically, the implementation extends to 4 virtual timesteps, which inevitably reduces the training and inference efficiency. However, the manuscript does not provide a detailed explanation or analysis of this trade-off, which would be beneficial for understanding the overall impact on efficiency.
2. The empirical comparison can be done in a more thoroughly by comparing with other latest state-of-the-art methods.
3. Some content in the paper seems unnecessary, such as Appendix A, which does not contribute significantly to the main arguments or findings and could be omitted for conciseness.

**Questions:**

Please see the weaknesses.

---

### Meta-Review · Area_Chair_stbR · 2024-12-19

**Metareview:**

This paper proposes the QSD-Transformer which is a quantized framework for spike-based Transformers by introducing a bi-level optimization strategy, incorporating information-enhanced LIF, and fine-grained distillation to rectify the attention distribution.  Most of the reviewers have positive comments on this work and part of them raised their score after the response. Thus, this paper can be accepted and please prepare the final version well.

**Additional Comments On Reviewer Discussion:**

Reviewer xmWz (Point Raised):
Concern over the potential reduction in training and inference efficiency due to the extension to 4 virtual timesteps.
Authors’ Response:
Clarified that the use of multi-bit pulses within a single time step during training (IE-LIF) and their conversion to binary pulses during inference reduces memory requirements and improves training speed. Provided data showing a 3.2× speedup in training and a 6.1× reduction in memory usage.

Reviewer xmWz (Point Raised):
Suggestion for a more thorough empirical comparison with other state-of-the-art methods.
Authors’ Response:
Included results comparing QSD-Transformer with the latest state-of-the-art method, QKformer, demonstrating the effectiveness of their approach.


Reviewer pkTr (Point Raised):
Lack of comparison with previous SNN and quantized ANN transformer models.
Authors’ Response:
Added comparisons with QKFormer and QAT-ViT models to strengthen the empirical evidence.

Reviewer pkTr (Point Raised):
Concern about the scalability of the method, given the use of small architectures for ImageNet experiments.
Authors’ Response:
Explained that the small architectures were the result of quantizing larger spike-based Transformer models and that the largest quantization baseline used was 55M, demonstrating scalability. Plans to apply the method to even larger models in the future were mentioned.


Reviewer pkTr (Point Raised):
Large training overhead due to multi-bit spikes and knowledge distillation.
Authors’ Response:
Provided analysis showing that IE-LIF and FGD techniques did not increase training overhead and in fact reduced it compared to traditional spike-based Transformers.

Reviewer pkTr (Point Raised):
Insufficient information in the transfer learning section, specifically about bit-width and accuracy without transfer learning.
Authors’ Response:
Clarified the bit-width used and provided accuracy results for CIFAR10/100 and CIFAR10-DVS with direct training, showing high accuracy.

Reviewer pkTr (Point Raised):
Request for firing rate information and changes in the self-attention part compared to the original Spike-driven Transformer-V2.
Authors’ Response:
Apologized for the oversight and presented the changes in the firing rates of the attention mechanism modules before and after quantization.


Reviewer xmWz:
Request for comparative experimental results with Fast-SNN.
Authors’ Response:
Agreed to add comparative experimental results with Fast-SNN, explaining how Fast-SNN inspired their IE-LIF approach, which uses multi-bit values during training and binary spikes during inference.

Reviewer xmWz :
Inquiry about the training efficiency compared to the original spiking transformer.
Authors’ Response:
Confirmed that the QSD-Transformer quantization approach improves training efficiency due to the use of IE-LIF neurons, which reduce training time and memory consumption.

Reviewer xmWz:
Feasibility of applying the method to NLP tasks.
Authors’ Response:

Confirmed the feasibility and added experiments on NLP tasks using the QSD-Transformer quantization framework, with comparisons based on Spikezip and SpikeBERT.


Reviewer pkTr :
Concern that IE-LIF neurons do not utilize temporal information, unsuitable for temporal benchmarks.
Authors’ Response:
Clarified that IE-LIF enables multi-time-step forward propagation during training, suitable for temporal benchmarks, and provided evidence of improved performance on CIFAR10-DVS.


Reviewer pkTr :
Limited comparison between ANN2SNN and Direct Training methods, with other methods outperforming QSD-Transformer.
Authors’ Response:
Addressed the concern by including a comparison with state-of-the-art ANN2SNN methods like SpikeZip and ECMT, and demonstrated the application of QSD-Transformer to models like Spikezip, showing improved results.



The authors effectively addressed the concerns and questions raised by the reviewers, providing additional experimental results and clarifications that strengthened the manuscript.
The decision to accept the paper was influenced by the authors’ ability to show improved training efficiency, the feasibility of applying their method to NLP tasks, and the suitability of their approach for temporal benchmarks.
The inclusion of comparative results with Fast-SNN and other state-of-the-art methods demonstrated the robustness of the QSD-Transformer quantization framework.
The decision also considered the overall contribution of the work to the field and the clarity of the responses to the reviewers’ points.

---

### Decision · Program_Chairs · 2025-01-22

Accept (Poster)